# Decreased Vertical Trunk Inclination Angle and Pelvic Inclination as the Result of Mid-High-Heeled Footwear on Static Posture Parameters in Asymptomatic Young Adult Women

**DOI:** 10.3390/ijerph16224556

**Published:** 2019-11-18

**Authors:** Jakub Michoński, Marcin Witkowski, Bożena Glinkowska, Robert Sitnik, Wojciech Glinkowski

**Affiliations:** 1Institute of Micromechanics and Photonics, Faculty of Mechatronics, Warsaw University of Technology, 02525 Warsaw, Poland; j.michonski@mchtr.pw.edu.pl (J.M.); m.witkowski@mchtr.pw.edu.pl (M.W.); r.sitnik@mchtr.pw.edu.pl (R.S.); 2Department of Sports and Physical Education, Medical University of Warsaw, 00581 Warsaw, Poland; boglinkowska@o2.pl; 3Centre of Excellence “TeleOrto” for Telediagnostics and Treatment of Disorders and Injuries of the Locomotor System, Medical University of Warsaw, 00581 Warsaw, Poland; 4Department of Medical Informatics and Telemedicine, Medical University of Warsaw, 00581 Warsaw, Poland; 5Polish Telemedicine and eHealth Society, 03728 Warsaw, Poland

**Keywords:** mid-high-heeled footwear, static posture parameters, young adult women, standing, spine curvatures, vertical trunk inclination

## Abstract

The influence of high-heel footwear on the lumbar lordosis angle, anterior pelvic tilt, and sacral tilt are inconsistently described in the literature. This study aimed to investigate the impact of medium-height heeled footwear on the static posture parameters of homogeneous young adult standing women. Heel geometry, data acquisition process, as well as data analysis and parameter extraction stage, were controlled. Seventy-six healthy young adult women with experience in wearing high-heeled shoes were enrolled. Data of fifty-three subjects were used for analysis due to exclusion criteria (scoliotic posture or missing measurement data). A custom structured light surface topography measurement system was used for posture parameters assessment. Three barefoot measurements were taken as a reference and tested for the reliability of the posture parameters. Two 30-degree wedges were used to imitate high-heel shoes to achieve a repeatable foot position. Our study confirmed the significant (*p* < 0.001) reduced vertical balance angle and pelvis inclination angle with large and medium-to-large effects, respectively, due to high-heel shoes. No significant differences were found in the kyphosis or lordosis angles. High-heeled shoes of medium height in young asymptomatic women can lead to a straightening effect associated with a reduced vertical balance angle and decreased pelvic inclination.

## 1. Introduction

The perception of a woman’s physical appearance by other people (both female and male) is significantly dependent on the type of footwear she is wearing. Wearing high-heeled shoes belongs to women’s social behavior that increases the attractiveness, elegance, and even self-confidence of women [1,2]. Over 50% of women wear high-heeled shoes for 1–8 h per day, and at least one-third of women in western countries wear high-heeled shoes daily [3]. A common belief about the adverse impact of footwear on the body has attracted the attention of many researchers to the problem of the impact of wearing high-heeled shoes on various biomechanical aspects of the body. In particular, it has been reported that the habitual wearing of high heels may result in static and dynamic changes in posture and spinal curvatures [4,5,6,7,8,9,10,11,12], as well as the lower extremities [13]. Studies also suggest that long-term wearing of high-heeled shoes may correlate with a higher rate of lower back pain [7].

High-heeled shoes may lead to the development of postural disorders of the head, spine, pelvis, and knees [14] that are characterized by head protraction [8], lumbar hyperlordosis [4,9,10,12,15] or decrease of the lordosis angle [6], unnecessarily increase the forward inclination of the body and its asymmetry [11], and also pelvic anteversion [4,5,9,15,16]. Additionally, an elevated risk of foot and ankle injuries from high-heeled shoes is reported [17]. The postural changes like a compensatory increase in lumbar lordosis and pelvic tilt are suggested to provoke lumbar back pain in habitual wearers of high heels [10,15,18]. Decreased lumbar lordosis was usually described in habitual wearers [14,19,20]. Increased lumbar lordosis [21] associated with high-heeled shoes has been reported for inexperienced wearers [9,22], or adolescent experienced wearers [15]. Cowley et al. [23] concluded in their review that increased lumbar lordosis angles were found predominantly in inexperienced users. Some authors suggest that high-heeled shoes do not affect lumbar lordosis [12]. Research results until now have been inconclusive with respect to which effects and to what extent they are caused by wearing high-heeled shoes. The inconsistency of the literature findings may be due to high-heel habituation grade, age of the users, small samples, a variation of investigated heel-height used in the experiments, and the variety of assessment methods used.

The low, medium, and high heel shoe categories are mentioned in the literature [24]. Dai et al. [9] used heels in range 50.3 ± 13.9. Drzał-Grabiec and Snela [11] conducted the study with two heel heights (4 cm and 8 cm). De Oliveira Pezzan et al. [15] used wooden shoes with 10 cm heel and 2 cm elevation in the metatarsal region. Schroeder and Hollander [4] used in their experiments different heel heights ranging from 7 to 11 cm.

Franklin et al. [14] conducted measurements with a three-dimensional electrogoniometer. Russell et al. [12] used a spinal mouse device to measure lumbar lordosis. Drzał-Grabiec and Snela [11] assessed the body posture parameters based on photogrammetry moiré pattern projection and fiducial markers attached to the skin. De Oliveira Pezzan et al. [15] also used fiducial markers but with single-camera photogrammetry. Dai et al. [9] analyzed standing left lateral radiographs, including that of the spine and pelvis, obtained in a standardized standing position barefoot and with high heels. Weitkunat et al. [8] used two standing lateral radiographs of the whole body taken using a biplanar low-dose radiographic imaging system (EOS Imaging, Paris, France). Comparing radiological methods and surface topography is a big challenge due to the differences in the imaged structures [25,26,27]. The unified interpretation of the results for various methods remains difficult. An additional barrier for the comparability of the results is the presence of moderate statistical relations of the studied parameters.

Recently, the reduction of radiation exposure has become a severe concern for researchers [28,29,30,31,32,33,34,35]. The most crucial argument in favor of the development of surface topography methods is the reduction of the cumulative X-ray effect during systematic diagnostic tests of patients with scoliosis [28,29,31,32,34,36,37,38,39,40,41]. There is a noted preference in the use of non-radiation dependent measurements [42,43,44,45,46,47,48,49,50,51,52,53]. However, even radiographs using a biplanar low-dose radiographic imaging system (EOS Imaging) may produce a total radiation dose as low as 0.1–0.7 mSv per participant [54].

Repeatability of the experiment also depends on the measurement uncertainty of the devices used to deliver image data and algorithms used to extract parameters and indices from that data. The majority of the studies do not take these factors into account. 

A more rigorous investigation is necessary to confirm the significant factors. The aim of this study was to investigate the effect of high-heeled footwear on static posture parameters in a homogenous sample of young adult women while standing. Surface topography (ST) was selected as the investigation tool to facilitate comparison of results, but also for its moderate availability, non-invasive character, and amount of delivered information about the body shape [18]. The experiment was performed while controlling all its core components: the geometry of the heel, the data acquisition process (including hardware), and the data analysis and parameter extraction stage. In this way, we intended to remove most of the uncontrolled variables, which should contribute to more reliable and repeatable results. All the steps are described in detail to provide a basis to reproduce the results in similar experiments in the future.

## 2. Material and Methods

### 2.1. Design

The experiment followed a repeated measures design, two conditions were examined, the initial state being barefoot, and the modified state wearing high-heeled shoes. The initial state was measured multiple times to test repeatability. No randomization was applied in order to follow the posture changes in each subject.

### 2.2. Subjects

A group of seventy-six healthy volunteers, students of the Physiotherapy Department of the Medical University in Warsaw, was initially enrolled in the study. Body mass index (BMI) was used to determine subjects at ranges reliable for surface topography measurements. The high body mass index may prevent the surface topography from detecting the severity of the curves. The reproducibility of surface topography has been found to be accurate in patients with BMI up to 29 [55]. The healthy normal BMI range (18.5–24.9) applies the same for men and women.

The inclusion criterion was at least two years of experience in wearing high-heeled shoes. The exclusion criterion was the occurrence of faulty posture, assessed by performed measurements [56]. Additional subjects with corrupted measurement data were removed from the study after performing the measurements. After the drop-out, a total number of 53 subjects were analyzed (Figure 1).

The study protocol adhered to the ethical standards of the Helsinki Declaration and the Institutional Review Board (IRB) of the Medical University of Warsaw approved the study (No. KB 158/2009, issued 25 August 2009).

### 2.3. Data Acquisition

The measurements were performed using a prototype custom-made structured light illumination (SLI) 3D scanner developed for the tasks of the project entitled “Telemedical, automatized system for three dimensional measurement, analysis, detection, monitoring and treatment of postural failures and deformations of the human body” in cooperation with the Institute of Micromechanics and Photonics (IMP), Warsaw University of Technology. The scanner was composed of a Casio XJ-A142 (https://www.casio-projectors.eu/euro/products/green-slim/xj-a142/) projector and a Point Grey FL2-08S2M-C (https://www.flir.com/products/firewire-cameras/?model=FL2-08S2M-C) industrial camera. The structured light method used was the 6-frame temporal phase-shifting (TPS) method, known for excellent accuracy and spatial resolution, with nine additional images encoding Gray code for fringe enumeration [43,57]. The camera calibration and projector-camera stereo-calibration used were from the 3DMADMAC solution [58].

The projector and the camera were synchronized using a photodiode, which made it possible to perform a single measurement in approximately 0.85 s. A glass plane surface of dimensions 1.2 m by 2.0 m with precise circular markers printed on an adhesive sheet and attached to the plane was used for system calibration and precision assessment. The measurement precision of a single point was 1 mm, and the spatial resolution was approximately 2–3 mm, depending on the position of the subject in the measurement volume. As the last step, the plumb line was calibrated to facilitate analysis of the body with respect to the gravity vector. The calibration was performed by measuring a loosely-hanging cylindrical artifact and fitting a cylinder to the resulting point cloud. The result of a measurement of a subject’s back was a point cloud consisting of approximately 100 thousand points, depending on the subject body size. The surface of the back was measured from a single direction, and the measurement was automatically aligned concerning the gravity vector using the plumb line calibration.

### 2.4. Application of Fiducial Markers

A set of anthropometric landmarks of the dorsum and shoulders was palpated on the surface of the back of a subject. A plain white, adhesive circular marker of 10 mm in diameter was attached to the skin in each palpated landmark location. The landmarks were:-C7—the spinous process of the vertebra prominens,-LAX/RAX—left and right axilla,-LSC/RSC—the inferior angle of the left and right scapula,-ThK—the apex of thoracic kyphosis,-ThK-LL—point of transition from thoracic kyphosis to lumbar lordosis,-LL—the apex of lumbar lordosis,-LVD/RVD—left and right Venus dimple,-IF—top of the intergluteal cleft.

The complete map of anthropometric points used in the experiment is presented in Figure 2. The marker at the transition from the thoracic kyphosis to the lumbar lordosis curve, used to calculate the thoracic kyphosis angle and lumbar lordosis angle, was chosen by the examiners based on their expert anatomical knowledge. The twelfth palpated rib was used to establish the position of the marker. A little more demanding to assess were the actual inflection point of the spinal curve and the kyphosis and lordosis angles. Points were found as the most prominent anthropometric point seen or palpated on the thoracic kyphosis and the deepest depression point of the lumbar lordosis. The correspondence of sagittal curvatures obtained on the point cloud and their radiograph equivalents could be considered best. However, these angles should be treated merely as values, generally describing the thoracic and lumbar curves of the spine. 

### 2.5. Barefoot Measurements

Barefoot measurements were performed at the beginning of every subject. The measurement was repeated three times to test the repeatability of the method. For each repetitive measurement, the subject would step out of the measurement volume, remain there for approximately 1 min doing simple movements, and then step back in. 

### 2.6. High-Heeled Measurements

For the high-heel measurement, instead of regular high-heeled shoes, wooden wedges with an angle of incidence of 30 degrees were used (Figure 3). The students were instructed to put the metatarsal heads on the floor and the rest of the foot on the wedge, to obtain an effect analogous to that of a mid-height high-heel shoe. 

The reasons for choosing the wedges over regular high-heeled shoes were multiple. Firstly, the wedge provides a constant angle of inclination of the foot and, as a result, a more homogeneous reaction of the body in all the subjects. Different stability characteristics of the shoes and degree of adaptation to the chosen pair of shoes could be ruled out. The angle of incidence for the wedges was chosen to match an average heel height. According to a survey conducted by the American Podiatric Medical Association, a heel of 2 inches (app. 5 cm) was reported as too high by 20% of women, whereas a heel of 3 inches (app. 7.5 cm) was reported as too high by 54% of women [59]. For a foot of average length, our setup would provide a heel of approximately 6–6.5 cm (2.5 inches) high, which should be in the acceptable range for most women. 

### 2.7. Data Analysis

Dedicated algorithms were developed to detect on-skin fiducial markers and calculate the parameters for evaluating changes in posture and balance according to current recommendations [42,60,61,62]. Analysis of the measurement data was performed using IMP’s proprietary FRAMES programming environment [63], an extensible framework used for different tasks involving point cloud data. It was written in C++ with minimal use of third-party programming libraries. The processing path consisted in:filtering measurement noise using FRAMES built-in algorithms,using a custom algorithm for automatic detection of the circular markers on the surface of the skin developed for this experiment, andmethods for estimating parameters connected with posture.

Filtering algorithms were used to remove parts of the measurement that did not represent the surface of the back, and remove measured surface edges, prone to noise because of the nature of the measurement technique.

The algorithm for marker detection was based on an analysis of intensity (grayscale value), as white markers were easy to distinguish from the surrounding skin. Since some over-exposed erroneous points appeared in the measurement, additional conditions had to be checked to ensure that a particular group of high-intensity points is circular, and its radius is within the expected values range. Only the positively verified point groups were regarded as marker areas. For each marker, its (x,y,z) center was calculated, and a textual name label was assigned based on the marker’s position on the body.

Some of the markers were not retrieved correctly using the automated method. Because the markers were partly reflective, the depth component within the area of the markers was noisier than in the rest of the measurement. The above was fixed by a dedicated semi-automatic algorithm, which used an initial marker position on the cloud pointed by the operator, segmented the surrounding area using a min-max ratio threshold, and calculated the center of the marker by fitting a 2D circle, disregarding the depth component. Then, the recovered center was projected back onto the cloud, obtaining the final (x,y,z) marker position. 

### 2.8. Postural Parameters Extraction

The markers LVD and RVD located at the left and right *fossae lumbales laterales* (“dimples of Venus”) were used to establish the base of the frontal plane measurement. Both the superficial topography indentations correspond to the upper parts of the sacroiliac joints. The transformation was found for each measurement independently by rotating the measurement space around the calibrated plumb line in such a way, that the Venus dimples were found in the frontal plane. Then, we used the markers located at vertebra prominens (C7), the point of transition from thoracic kyphosis (ThK) to lumbar lordosis (LL) (ThK-LL) and top of the intergluteal furrow (IF) to calculate the parameters for monitoring posture change. The postural parameters included:-vertical balance angle (vertical trunk inclination in the sagittal plane)(VBA), which is the angle between the vertical axis and the line connecting C7 and IF, the positive value corresponds to the subject leaning forward,-thoracic kyphosis angle (TKA), which is the angle between normal vectors of planes fitted in the area around points C7 and ThK-LL, with an additional offset to C7 equal to 5 mm in the direction of ThK-LL,-lumbar lordosis angle (LLA), which is the angle between normal vectors of planes fitted in the area around points IF and ThK-LL, with an additional offset to IF equal to 15 mm in the direction of ThK-LL,-pelvic inclination angle (PIA), which is the angle between the normal vector of the plane fitted to the area between LVD and RVD and the horizontal axis.

All the vectors were projected onto the sagittal plane before calculating the angles. Additional markers at the left and right scapula (LSC, RSC), left and right axilla (LAX, RAX) and the deepest point of lumbar lordosis (LL) were used to calculate the parameter for determining possible faulty posture:-the Suzuki Hump Sum (SHS) [64,65,66], calculated as the sum of the difference in depth between points of contact of lines in the axial plane tangent with the surface of the back, on three different levels: halfway between LSC and RSC, at ThK-LL and LL.

The principles of extracting all the parameters are presented in Figure 4. Out of the angles used for monitoring posture change, calculated for the three measurements without heels for each subject, the average value was calculated and used as the reference. The exclusion criterion was based on the SHS parameter. Although this parameter is well established in Surface Topography (ST) measurements [26,27,60,65,66,67,68], literature does not provide specific cut-off values for faulty posture. Thus, subjects with outlier values of the SHS distribution in any of the measurements were suspected of having a faulty posture and were excluded from the study. This parameter was used only as a base for exclusion and was not used in further analysis.

### 2.9. Influence of Marker Placement Accuracy

The position of attached fiducial markers affects directly the calculated parameters. Additional analysis was performed to test the influence of their position on the obtained results. The procedure was as follows:for each subject, for each landmark, draw a random displacement vector from the N(0, 10) distribution, each vector component was drawn independently,after applying the displacement vector to each landmark, the point was projected to the cloud to find a real point on the surface.

The random displacement vector was constant for each subject. This is equivalent to a different marker placement throughout the whole session. The values of parameters were calculated again for the new set of markers, and the results were statistically analyzed. The procedure was repeated ten times and was performed for parameters, which were statistically significant for the original marker positions. Three runs of application of marker displacement vectors for one measurement is shown in Figure 5. 

### 2.10. Statistical Analysis

Then, R version 3.5.1 (The R Foundation for Statistical Computing, Indianapolis, IN, USA) software was used to compare the distribution of the angles measured with and without heels. Outliers were identified as outside 1.5 The interquartile range (IQR). The measurements that qualified as outliers were examined for correctness. The Wilcoxon signed rank test was used to search for statistically significant differences between the two subject states (barefoot and heels) for any of the monitored angles and to reveal whether mean barefoot position measurements are statistically significantly different from high-heeled position measurements.

The effect size was assessed using the matched pairs binomial correlation coefficient, where values less than 0.3 were considered *small*, between 0.3 and 0.7—*medium*, and more than 0.7—*large*. Repeatability of barefoot measurements was tested using the Intraclass Correlation Coefficient (ICC3,3) with a mean-rating (*k* = 3), absolute-agreement, 2-way mixed-effects model. The result was interpreted according to the guidelines for ICC inter-rater agreement measures, where: Less than 0.40—poor; Between 0.40 and 0.59—fair; Between 0.60 and 0.74—good; Between 0.75 and 1.00—excellent.

## 3. Results

During the 3D data pre-processing stage, a complete set of markers required for further analysis could be successfully extracted in 56 subjects. The rest was removed from the dataset. The high drop-out rate at this stage was caused by the poor quality of the fiducial markers. Three subjects had outlier values of SHS and were excluded from the study. Then, five measurements selected as outliers in the vertical balance angle distribution, and four in the pelvic inclination angle distribution were found, nine in total. Distributions of thoracic kyphosis angle and lumbar lordosis angle did not show any outliers. All outliers were examined for any factors that would render them unacceptable for the experiment: faulty posture, measurement and calculation errors. No irregularities were found, and the measurements were kept in the dataset. 

After this procedure, 53 complete measurements were left for analysis. The remaining subjects were 20.4 ± 1.2 years old, characterized by BMI 20.2 ± 2.2. The summary of the calculated angle distributions is presented in Table 1. Intraclass Correlation Coefficient (ICC3,3) for barefoot measurements are presented in Table 2. Significant differences between the barefoot and high heel conditions were found in the vertical balance and pelvic inclination angles for *p* < 0.001 using the Wilcoxon signed-rank test. The thoracic kyphosis and lumbar lordosis angles did not differ significantly. All results of the effect size for the vertical balance angle, the pelvic inclination angle, for thoracic kyphosis angle, and lumbar lordosis angle—*small* are presented in Table 3.

The influence of fiducial marker placement was examined for the pelvic inclination and vertical balance angles, where the difference was statistically significant, and the effect size at least small. Ten repetitions of the random movement of the fiducial markers on the measured surface were performed. The displacement of markers was 13.4 ± 7.2 mm. The W statistic for the vertical balance angle was 67 ± 21 with a *p*-value less than 0.001 for all random runs and 215 ± 18 for the pelvic inclination angle with a *p*-value less than 0.001 for all random runs. The effect size was −0.90 ± 0.03 in the vertical balance angle (*large* effect size in all cases), and −0.69 ± 0.02 in the pelvic inclination angle (*medium* to *large* effect size in all cases).

## 4. Discussion

There is a lack of consensus in the literature regarding the posture changes caused by the use of high heel shoes due to several factors. This study aimed to investigate how medium-high heeled footwear influences static posture parameters. The majority of the symptoms associated with wearing high-heeled shoes are considered attributable to the observed adaptive biomechanical phenomena. 

No significant effects on the lumbar lordosis angle under static conditions found by Schroeder and Hollander [4] were confirmed in this study. A small reduced pelvic tilt was similarly present. Our study did not confirm any moderately increased transversal pelvic rotation. Other postural effects of high-heeled shoes were mentioned in the literature, namely, head protraction [8], postural disorders of the head and spine [14], pelvic anteversion [4,5,9,15,16], and knee valgus [8,10,16]. Only the selected parameters were evaluated in our study. Head protraction and knee position were not evaluated in the present study. The debate remains open about the potential impact of high heels on posture and lumbar hyperlordosis [4,9,10,12,15] or its decrease [6], pelvic tilt changes and whole-body sagittal balance. Silva et al. [10] reviewed studies searched in the Scopus, SciELO, and PubMed databases between 1980 and 2011 regarding the effects of high heeled shoes on the body posture of adolescents. They suggested that wearing regularly high heels can lead to permanent malposition of the spine and the legs. The results of our study show the wearing mid-high-heeled shoes may decrease the forward inclination of the body [11], and pelvic anteversions [15]. However, asymmetry [11] while wearing high-heels was not found. The effect of producing an uneconomic body position [8,9] could not be assessed using our methods.

Postural changes like a compensatory increase in lumbar lordosis and pelvic tilt are suggested to provoke lumbar back pain in habitual wearers of high heels [10,15,18]. Decreased lumbar lordosis was usually described in habitual wearers [14,19,20]. Increased lumbar lordosis [21] associated with high-heeled shoes has been reported for inexperienced wearers [9,22], or adolescent experienced wearers [15]. Cowley et al. [23] concluded in their review that increased lumbar lordosis angles were found predominantly in inexperienced users. Some authors have suggested that high-heeled shoes may not affect lumbar lordosis [12]. This is a duplicate from the introduction 

Similarly to the results obtained in the current research, the angle of the forward trunk inclination was found to be statistically significant. Its increase gradually with the increasing height of the heels [11] was already suggested in other studies [11]. The authors explained observed changes as the reaction in the body’s center of gravity and the attempt to maintain postural stability. The only significant difference was noticed between barefoot patients and those wearing 4-cm high-heeled shoes. When measuring the parameters in patients wearing 10-cm heels, the trend toward increasing body inclination was preserved; however, these differences were not significant. The paper mentions that higher values were observed for the angle of trunk inclination, despite the data showing a decrease in the angle of trunk bend (ATB angle), rendering the conclusions unreliable.

The cervical spine may show increased lordosis due to the forward displacement of the head [18]. Weitkunat et al. [8] found that most of the high heels-related adaptive responses to the antero-cranial shift of the center of gravity occur in the lower extremities, especially the knees. An additional mechanism to shift the center of gravity backward was an increase of cervical lordosis. Franklin et al. [14] showed significantly lower anterior pelvic tilt, lumbar lordosis, and sacral base angles with high heels when compared with zero heel inclination using a three-dimensional electrogoniometer. Russell et al. [12] using a spinal mouse device showed that high-heeled shoes did not affect lumbar lordosis. Drzał-Grabiec and Snela [11] using moiré photogrammetry found that wearing high-heeled shoes increases the forward inclination of the body and increases its asymmetry. However, the results are unreliable due to the absence of parameter definitions used to conclude, as well as discrepancies between the presented data and the text. De Oliveira Pezzan et al. [15] used custom-built software for postural assessment and photographs in the sagittal plane and extracted angles based on the location of fiducial markers found that the effect of increased lumbar lordosis and pelvic anteversions. Schroeder and Hollander [4] showed a small to moderate the effect of high-heeled shoes on static and dynamic pelvic positions (sagittal pelvic tilt and axial pelvic rotation) in females habituated to the use of high-heeled footwear. Additionally, they found no effects of footwear on the static or dynamic lumbar lordosis [4].

Spino-pelvic radiographic parameters characterizing sagittal balance are clinically relevant [8,38,69,70,71,72]. Dai et al. [9] analyzing standing left lateral radiographs of the spine and pelvis revealed increased lumbar lordosis. No significant differences for sagittal balance parameters (SS, PT or PT/SS) between the barefoot and high-heel positions were found. One study [9] showed that the radiographic sagittal vertical axis (SVA) was always positive and was worse after wearing a variety of heights of high-heel shoes. Finally, they concluded that the SVA was significantly more extensive under the 45.5 mm height high heel use than barefoot. Weitkunat et al. [8] studied biplanar standing lateral radiographs of the whole body found in some cases, increased cervical lordosis. Substantial correlations and pronounced differences between the barefoot/high-heeled conditions were found in the C7 sagittal vertical axis (SVA), the cervical lordosis, the knee flexion angle, and the femoral obliquity angle. No statistically significant changes were seen for thoracic kyphosis, lumbar lordosis, or the measures of pelvic sagittal inclination. A radiographic study by Aota et al. [73] found that arms relaxed in front with hands loosely clasped produce the least negative shift in SVA is the best arm position for SVA measurement.

Authors of the radiographic studies used the images of patients who were positioned with fingers on the clavicles that could significantly influence the angle of lordosis [74]. The surface topography study using structured light has shown that position with fingers on clavicles does not influence vertical trunk inclination and kyphosis, but significant changes of the lordosis angle were found [74]. The positioning of the body during the posture assessment may influence the spinal curvatures [73,74,75,76] and sagittal alignment.

Out of all the methods used to examine static changes in posture, rasterstereography (also referred to as surface topography, ST) offers the most reliable information [46,47,77,78,79,80,81], while not exposing the measured persons to ionizing radiation. Various parameters and indexes are drawn from the surface-topography and radiographic data that may confuse the interpretation of the results. Surface topography is usually used to detect or monitor scoliosis deformities [25,42,49,57,67,82,83,84,85,86,87,88,89,90,91,92,93,94,95,96], few publications address the physiological curvatures of the spine, lumbar lordosis and thoracic kyphosis [26,42,48,64,82,97,98,99,100]. Radiographic measurements of SVA, PT, LL, SS, or PI are based on radiographic points assessed on the full spine radiograms [72], which are inaccessible in surface measurements. Parameters extracted by means of recently developed non-invasive optical systems (including moiré projection method and surface topography) are frequently novel and specific for each device [12,14,47,82,88,99,101,102,103]. Some of the indexes are recommended by the SOSORT [60]. However, the guidelines are known mostly to researchers dealing with scoliosis, and not applied during most experiments.

In our study, significant differences were found for the vertical balance angle and pelvic inclination angle. Unexpectedly Weitkunat et al. [8] found no statistically significant changes for thoracic kyphosis, lumbar lordosis, and the measures of pelvic sagittal inclination. The absence of statistically significant changes of the lumbar lordosis was explained by the high variability in pelvic incidence in the study sample.

We assume that vertical trunk inclination in the sagittal plane may express the same condition of the vertebral column in surface topography as SVA in lateral view radiography. However, the actual relation between these parameters has not been investigated.

### 4.1. The Biomechanical Aspect of Results/Observations

Biomechanical phenomena described in particular publications may lead to generalized and explainable impressions. Many studies on the influence of high heels on the sagittal balance focused on these parameters due to the possibility of an antero-cranial shift of the body’s center of gravity while standing on high-heeled shoes [8,9,104]. The cranial and anterior shift of the center of gravity [8,10,72,105] of the body when standing in high-heeled shoes was described in the literature. Lee et al. [18] showed a compensatory posterior tilt of the whole upper body with high heels. When standing in high-heeled shoes, the body’s center of gravity is being shifted cranially and anteriorly [18,104]. Weitkunat et al. [8] observed that wearing high heels led to increased flexion of the knees and more ankle flexion or to increased cervical lordosis to compensate for the antero-cranial shift of the body’s center of gravity. Increased lumbar lordosis and changing body position to an uneconomic [9] or permanent malposition of the spine and the legs [10] was described due to high-heeled shoes. However, in the literature, high-heeled shoes were not always found as the factor that increases the forward inclination of the body, its asymmetry [11], and pelvic anteversions [15] that may lead to an uneconomic body position [8,9]. Some studies [1,9,20,106] have suggested that high heels induce a vertical integration in the sacrum, pelvic tilt and lumbar adjustment due to increased activity of the hamstring muscles to counteract the abnormal gravity line. A small to moderate effect of high-heeled shoes was observed on static and dynamic pelvic positions (sagittal pelvic tilt and axial pelvic rotation) in females habituated to the use of high-heeled footwear [4]. The wearing of high-heeled shoes may cause a “chain reaction” of postural alteration superior to the ankle, where the pelvis, rather than the lumbar spine that is involved in postural compensatory strategies [4]. The statistically significant gradual increase of the trunk forward inclination angle with the increasing height of the heels [11] was explained as the reaction in the body’s center of gravity and the attempt to maintain postural stability. The cervical spine may show increased lordosis due to the forward displacement of the head [18] as an additional mechanism to shift the center of gravity backward. Most of the high heels related adaptive responses to the antero-cranial shift of the center of gravity occur in the lower extremity, especially the knees [8].

### 4.2. Influence of Precision of Fiducial Marker Placement

The markers were placed once for all the measurements, did not change their position on the skin, and thus provided a good base for comparison. Additional tests were conducted for the statistically significant parameters that showed at least a small effect size: the vertical balance and pelvic incidence angles. Tests were performed according to the described procedure. 

The pelvic incidence angle and vertical balance angle can be considered reliable measurements. All ten random tests showed statistically significant results for the parameters, and the effect size did not decline. 

In the case of pelvic incidence angle, the area around the left and right dimple of Venus markers used for calculation of the angle was large and was related closely to the orientation of the pelvis. For the vertical balance angle, the distance between the two markers used for calculation was considerable compared to possible inaccuracy in placing the markers. Additionally, possible inaccuracy in the frontal plane was small due to the location of the markers along the spine line, and the inaccuracy vector in the sagittal plane was almost parallel to the line connecting the two markers, thus did not have much influence on the angle value. 

### 4.3. Limitations of the Study

The general limitations of this study concern the acquisition of the 3D image that can only document a short moment of a body’s position. However, this limitation appears equally in static examinations regardless of the measurement method used for image acquisition: radiogram, moiré photogrammetry, rasterstereography, 3D structured light. None of the methods adjust to the dynamic process of balancing. The repeatability of three subsequent barefoot measurements was high, which implies that the captured differences between the barefoot and high-heeled conditions were significant at that moment in time. 

A limitation of this study is also the inability to measure the surface of the nude skin of the dorsum up to the hairline, which usually makes the skin of the occiput unavailable for the measurement. The comparison of the cervical lordosis studies with surface topography is not possible until the bald occipital protuberance can be exposed to the structured light. 

The limitations of comparison of the radiographic with surface topography studies are concerned in the measures and indexes [88,103,107]. The only trend of observed changes can be compared to some extent. Radiographic measurements of SVA, PT, LL, SS, or PI are based on radiographic points that are not achievable by surface topography and vice versa. Surface topography addresses anthropometric points that are seen or palpable superficially on the patient’s body [25,103]. The measurements were performed from the glutei upward, and only the posterior part of the torso was acquired. This setup impeded us from considering knee and ankle flexion, where additional compensations could occur, as reported by other studies [8,104]. 

## 5. Conclusions

In our study, significant differences were found for the vertical balance angle and pelvic inclination angle, which correspond to a decrease in the forward inclination of the body and pelvic anteversion.

The strength of the study was the well-delineated model and measurement methodology of the 30 degrees inclination high heel. The parameters measured in surface topography and radiography require cross-calibration in future research that may improve the understanding of the postural relations. 

There are reasonably clear premises in the literature about the differences in effects caused by high heels depending on their height. In the model used in our research, the heel height is classified as an average, and it is probably the reason for the obtained results. 

It seems that the problem of the impact of the heel height still leaves many questions. Subsequent research should include, among others, trunk examination of 360 degrees and dynamic tests. The current results can be applied to people working statically in heels for many hours. A comparison of the impact of different heel heights on the postural condition should also be included in the study plan. With particular interest should be addressed the question of whether one can adjust the heel height, which will be optimal for body posture and curvature of the spine? The research is based on the mechanisms of symptom formation in orthopedics. In the next step, there is a particular need to standardize the methodology of surface topography research and to obtain a reference system for radiological studies, which will increase the reliability and repeatability, especially in the sagittal alignment tests. The constant angle of heel elevation can be used in future studies focusing on the knee and hip flexion effect. 

## Figures and Tables

**Figure 1 ijerph-16-04556-f001:**

The procedure of selecting subjects for the final trial. SHS: Suzuki’s Hump Sum.

**Figure 2 ijerph-16-04556-f002:**
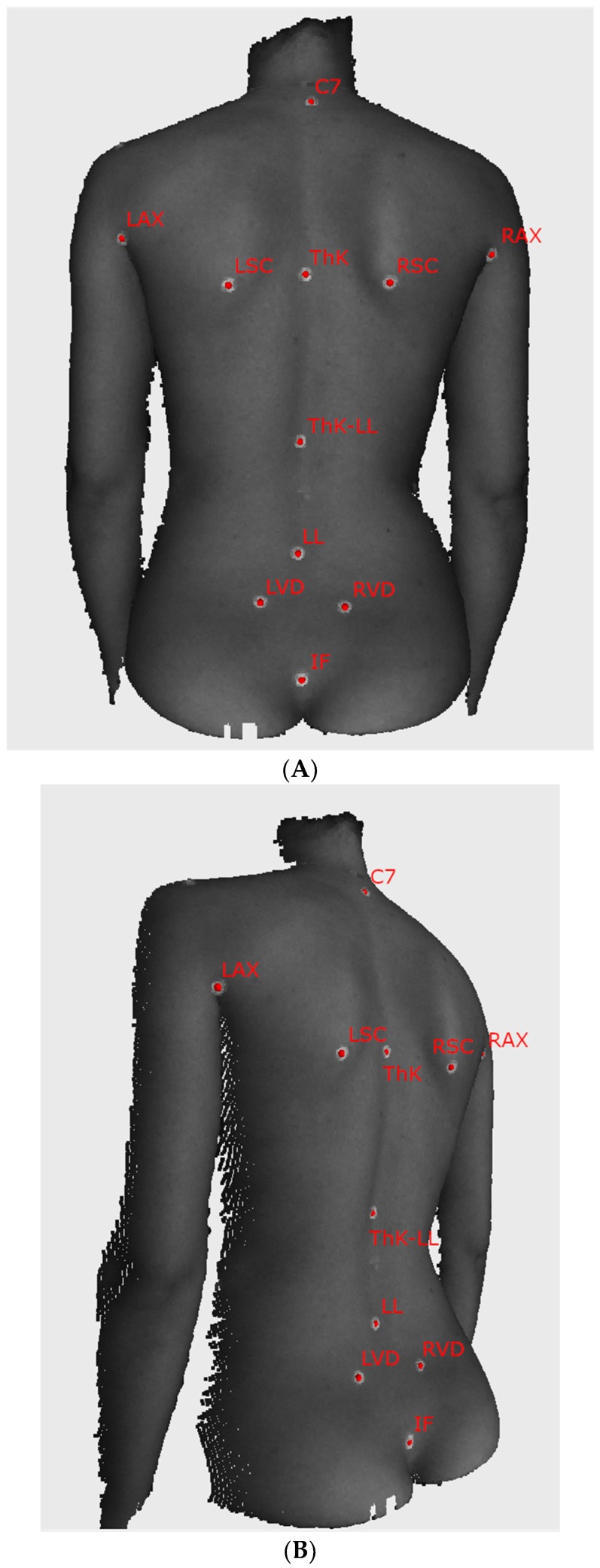
The sample images of the examined subject with self-adhesive markers on the skin ((**A**) posterior view, (**B**) oblique view). Markers used in the analysis are captioned with the name of the corresponding anatomical structure. C7—the spinous process of the vertebra prominens, LAX/RAX—left and right axilla, LSC/RSC—the inferior angle of the left and right scapula, ThK—the apex of thoracic kyphosis, ThK-LL—point of transition from thoracic kyphosis to lumbar lordosis, LL—the apex of lumbar lordosis, LVD/RVD—left and right Venus dimple, IF—top of the intergluteal cleft.

**Figure 3 ijerph-16-04556-f003:**
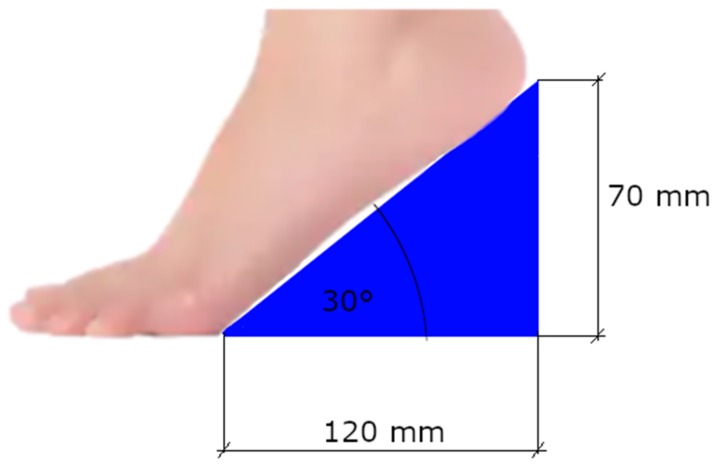
Schematic view of the wedge used as the heel with dimensions. The maximum height of the wedge was slightly larger than the heel effect it had on the foot.

**Figure 4 ijerph-16-04556-f004:**
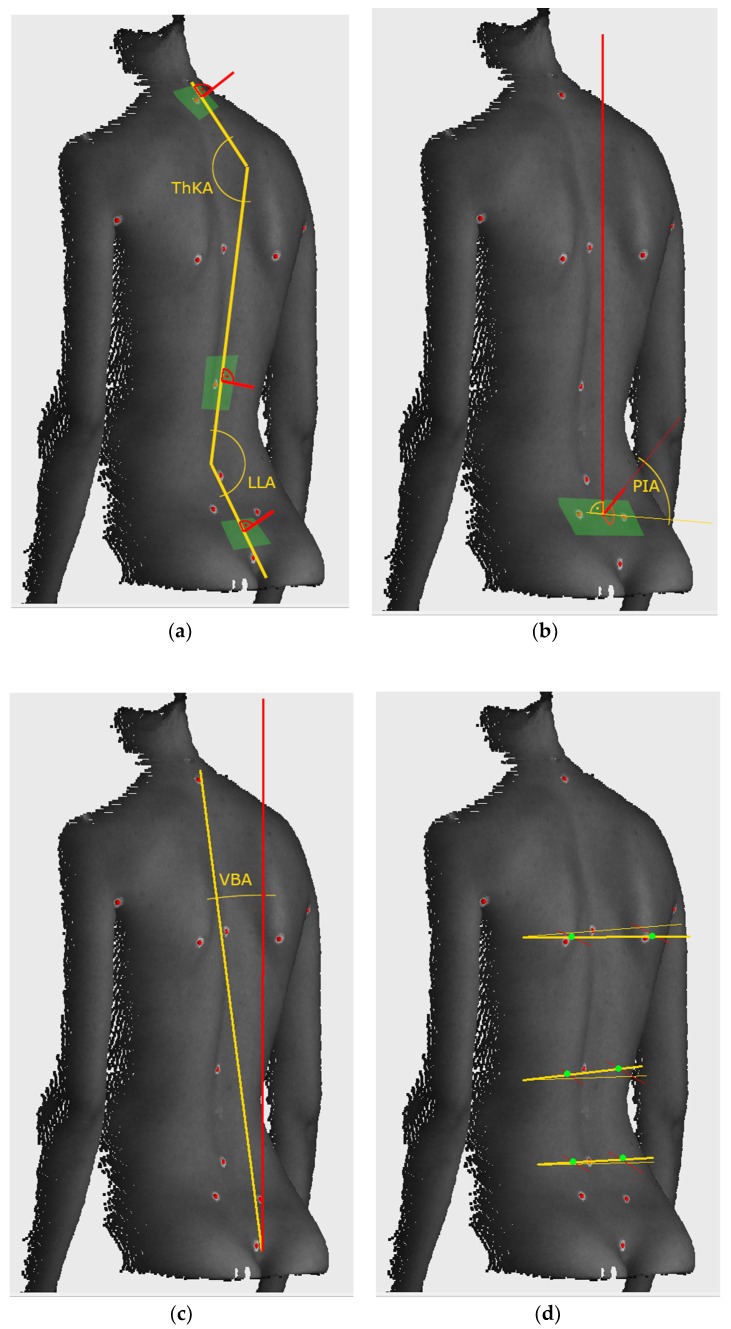
Calculation principle of parameters: (**a**) Cobb angles, (**b**) pelvis inclination, (**c**) vertical balance angle, (**d**) components of SHS. Green areas denote approximate parts of point cloud used for calculation. THKA: thoracic kyphosis angle, LLA: lumbar lordosis angle, PIA: pelvic inclination angle, VBA: vertical balance angle (vertical trunk inclination in the sagittal plane.

**Figure 5 ijerph-16-04556-f005:**
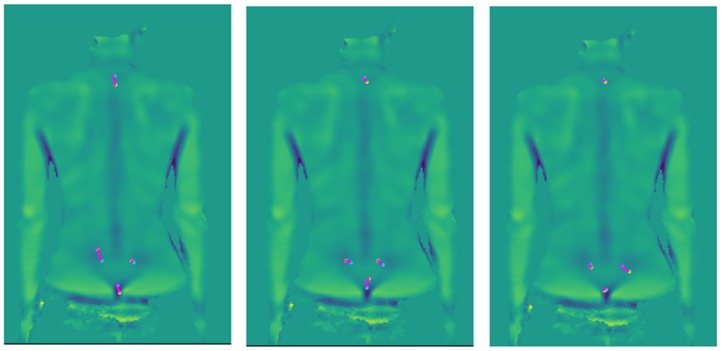
Example of artificial marker displacements. The blue point is the original marker position, and the yellow point is the marker after the displacement, and in magenta is the displacement vector.

**Table 1 ijerph-16-04556-t001:** Summary of distributions for all the angles.

Parameter	Distribution	Minimum [°]	Lower-Hinge [°]	Median [°]	Upper-Hinge [°]	Maximum [°]
Thoracic kyphosis angle	Barefoot	11.5	24.8	32.5	35.5	47.0
	High Heels	12.9	25.3	31.1	36.8	47.7
Lumbar lordosis angle	Barefoot	15.1	28.7	33.8	40.5	50.6
	High Heels	14.0	26.4	32.6	39.0	50.0
Pelvic inclination angle	Barefoot	16.5	25.2	28.5	31.8	37.6
	High Heels	15.1	23.0	26.8	31.5	36.6
Vertical balance angle	Barefoot	0.0	2.6	3.5	5.3	10.0
	High Heels	−1.1	1.2	2.4	4.0	9.7

**Table 2 ijerph-16-04556-t002:** Intraclass Correlation Coefficient (ICC_3,3_) for barefoot measurements.

Parameter	ICC_3,3_
Thoracic kyphosis angle	0.94
Lumbar lordosis angle	0.99
Pelvic inclination angle	0.96
Vertical balance angle	0.98

**Table 3 ijerph-16-04556-t003:** Wilcoxon signed rank test results and effect size.

Parameter	Wilcoxon Signed Rank Test (Heels vs. Barefoot, Paired)	Matched Pairs Rank-Biserial Correlation
Thoracic kyphosis angle	W = 547, *p* = 0.13	0.23 (*small*)
Lumbar lordosis angle	W = 620, *p* = 0.40	−0.13 (*small*)
Pelvic inclination angle ***	W = 201, *p* < 0.001	−0.72 (*medium* to *large*)
Vertical balance angle ***	W = 51, *p* < 0.001	−0.93 (*large*)

*** *p* < 0.001.

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
