# Peer review of "Decreased Vertical Trunk Inclination Angle and Pelvic Inclination as the Result of Mid-High-Heeled Footwear on Static Posture Parameters in Asymptomatic Young Adult Women"

_ijerph, 2019, doi:10.3390/ijerph16224556_

Round 1
Reviewer 1 Report
The height of the heels is a question often asked to the doctor of the spine. The majority of health articles that patients read highlight significant static changes with high heels.
The popular image of excessive pelvic anteversion and lordosis is confirmed by Lee. But the confusion probably stems from the lordotic visual effect of the contracture of the gluteal muscles as pointed out by Mosner and Bryan. In fact Weitkunat notes no change in radiological parameters of the spine.
At age 20, and in the absence of pathology, the anterior projection of the center of gravity well described by Snow and Williams is rebalanced by a pelvic retroversion which automatically leads to hypolordosis. Mika notes that this lumbo-pelvic compensation decreases with age.
These compensations are also modified with the wearing time of the high heels (Santos). One of the weaknesses of the study comes from the fact that we do not know if a postural reflex is recorded or if it is a real modification of the posture.
In conclusion, this work confirms that all compensations are not limited to the lower limbs and that there are lombo-pelvic changes even at 20 years.
Minor revisions
Can we know the proportion of men and women and the time of adaptation to the posture?
Author Response
Dear Reviewer,
Thank you for your comments.
We applied small corrections accordingly to your suggestions.
In detail, we do agree that static changes seem to be most highlighted for patients. Thank you for mentioning observations by Lee, Mosner, and Bryan, Weitkunat, Snow and Williams, Mika, and Santos. The new paragraph was added about the reliability of the study. The highly significant ICC may serve as the confirmation of how wearing high heels rather modify the posture than the reflex.
Indeed, and we found that the trunk responds to the vertical location of the heel bones, no lower limbs adaptation was examined in our study.
No males were examined in this study so that no conclusion can be drawn about the posture adaptation differences.
Kind regards
Sincerely
Authors
Reviewer 2 Report
Dear Editors,
Dear Authors,
I would like to thank you for the opportunity to review the submitted manuscript:
„The decreased vertical trunk inclination angle and decreased pelvic inclination in asymptomatic young adult women as the result of mid-high-heeled footwear on the static posture parameters”.
The authors reported a well conducted study describing effects of mimicked high-heels (approx. 7 cm) on postural parameters of the spine in the sagittal plane assessed with a relatively novel device providing an optical back surface reconstruction, somehow comparable to well-established rasterstereography devices like the Formetric® system. The authors aimed at reproducing earlier results from Schroeder & Hollander (2018) describing high-heeled shoe effects on the lumbar lordosis and the pelvis inclination under static conditions, but with a higher degree of study conditions’ standardization in order to boost the evidence of the earlier rasterstereography findings. The authors reported a huge amount of details describing the novel working principles of their assessment tool. They found a decrease of upper body forward inclination and a pelvic anteversion, while thoracic kyphosis and lumbar lordosis remained unchanged.
I appreciate very much that the authors made an approach to replicate earlier findings, as this might strengthen earlier results obtained with a surface topography device, while the whole body of the literature appears to be still inconsistent when the effects on spine shape or posture are summarized – probably due to multiple factors, e.g. x-ray vs. back surface reconstruction, the heel-height, the age of the wearers and the degree of high-heel habituation.
Nevertheless, there are some recommendations that should be addresses before publishing.
Please let me start with some minor recommendations:
In the results section (starting from line 212) in Fig. 4, the labelling of the y-axis is missing. In Table 1, the units of your dependent variables are missing, e.g. [°] for all angles. Please reconsider the number of decimals you choose to report the results. Is it appropriate to report the back shape angles with an accuracy of 0,01° (Table 1)? And F-values or p-values should be limited to 3 decimals (Tables 2 and 3).
Then, please let me make some suggestions to clarify the reported ‘subjects’ drop-out and your statistical approach:
In the methods section (starting from line 67), you report that 76 young adult females were enrolled in the study. Then, you give the average age and BMI, and then you describe a drop-out to reasons. Does the given age/BMI refer to the sample analyzed later in your statistical calculation, or to the total of the initially recruited persons? To my opinion, you should report the sample characteristics later, when the drop-out was shown in order to describe the remaining sample. Another recommendation is also referring to the drop-out you described markedly later (in the results section from line 213). You started with 76 persons, 2 SHS outliers were removed, 9 VBA and PIA outliers were removed, but only 43 complete measures were left for analysis (line 219): 76 – 11 = 65 ≠ 43 ? You have to clarify this discrepancy – I would suggest another figure, preferably a flow chart showing all steps of drop-out for any reasons. If another figure is not allowed, you should reconsider your Fig. 4, which might be redundant as it gives no more information than Table 1.
Referring to your statistical approach I have to address some questions or recommendations:
You reported that the barefoot test was repeated three times to test repeatability (line 118), but you did not report any reliability coefficient or standard error! Please give some numbers, probably an ICC1,3 and the standard error of the mean of the repeated measures (e.g. SEM = SD / √n referring to the 3 individual’s repeated barefoot measures with averaging afterwards for the whole sample, see Schroeder et al. (2015), reliability analyses of rasterstereography parameters, Eur Spine J). You demonstrated in detail that your data were normally distributed (line 220 and Table 2), but you used non-parametric statistics (median, 25% and 75% quartiles) to describe the results (Table 1 and Fig. 4), and then you calculated parametric procedures to test significance (line 221 and Table 3). You should explain this, or give a rebuttal! It may be preferable to use the interquartile range with minimum and maximum to describe the sample’s variation, but if you use an ANOVA it should be rather the mean ±SD (probably accompanied by minimum and maximum). Moreover, you averaged the three repeated barefoot measurements, and this mean was used as a reference value for comparisons with the mimicked high-heel condition (line 196). This may improve the reliability of the barefoot condition values, but what about the mimicked high-heel condition values that were taken only once? I suggest that the single measures may be suitable, when you give some numbers demonstrating absolute and relative reliability (see above: SEM and ICC). But when you compare only one barefoot value and one mimicked high-heel condition value, there is no need to calculate a 1-way ANOVA for repeated measures. You should choose paired t-tests for significance testing, and the accompanying effect size Cohen’s dz. Using an ANOVA implies that you analyze controlled effects, which in fact was not the case (you tried to replicate the findings of Schroeder & Hollander (2018), who conducted a repeated measures study with a baseline test and two following shoe conditions in a randomized order: flat vs. high-heeled, but you chose a simple “pre-post” design – first the reference barefoot and then the intervention ‘mimicked high-heels’).
As a major concern, I have to state that the structure of the manuscript appears to be a little difficult to read. In the following I will try to make some recommendations:
Abstract:
Structure is OK with one exception: “Seventy-six […] were enrolled” (line 18). And “Forty-three measures were obtained for analysis” (line 24). Please put it together. For instance: ‘After a drop-out for reasons, 43 of 76 participants were analyzed.’
Introduction:
Back ground relevance of the topic was well described (line 36 – 45), and the study goal was given clearly (line 59 – 66).
But in between the rationale of the study idea and the description of the remaining research gap is not quite clear (46 – 50).
And the paragraph reporting disadvantages of x-ray imaging and non-invasive alternatives (line 51 – 58) appears to be lying outside of the context, so far. I suggest some reorganization of these paragraphs.
Proposal:
Probably, it may help, if you make a clear statement of the “inconsistency of the literature findings” and mention probably underlying reasons like ‘variation of investigated heel-height’ or ‘used assessment tools’ or ‘high-heel habituation grade and age of the users’. This would be possible, when you use some shortened paragraphs that are positioned so far in the discussion section.
To my opinion, the paragraph ‘Height of the heels’ (line 243…) is not appropriate for the discussion, because no spine shape effects are reported, but it can be shortened and then used in the introductions rationale to describe the variety of used heel-heights in the respective literature.
Furthermore, the paragraph ‘Method of measurements and its safety’ does not report any findings that can be compared to the present study’s results at the actually given state. Thus, it is not useful for the discussion under its present form, but it can be shortened and then used also in the introduction to underline the variety of methods also affecting the varying high-heel shod standing posture findings.
This would open the door for the paragraph describing x-ray disadvantages and safety advantages of back surface reconstruction tools.
Material and Methods:
I would appreciate a structured methods section (design, sample, assessment tool with outcome variables, data processing and statistics), if the journal allows subheadings. Under its present form, the methods section is not very comprehensive, to my opinion.
For instance, it is not easy to find information reporting clearly the chosen study design (e.g. repeated measures under two conditions in the sense of a pre-post observation without a randomized order). A subjects section should give all the information needed to understand how many persons were analyzed. In contrast, the first paragraph gives some details about the subjects initially enrolled, and then reports inclusion and exclusion criteria (line 68 – 80), but there are no clear statements about the resulting sample, here. This can be provided by the aforementioned recommendation of a flow-chart that clearly points out that the remaining sample consisted of 43 of initially 76 females, and that there was a drop-out for reasons (2 faulty posture, 9 statistical outliers in different variables leading to the removal of participants in any variable sample).Discussion:
I would have expected that the discussion starts with a brief repetition of the study goal and the central findings in order to discuss the own results facing the respective literature.
In contrast, the authors start with a paragraph (line 239 – 241) that would rather be suitable for the conclusions beginning, to my opinion.
The following paragraphs ‘Height of the heels’ and ‘Method of measurements and its safety’ are not opposed to the own data. High-heel effects on any postural parameter are not mentioned, here; solely, the general impact of varying heel-height and assessment tools on any results. Thus, these paragraphs (in a shortened version) are probably helpful in the introduction to support the rationale of the study idea.
Why do the authors not start with their results in comparison to Schroeder and Hollander (2018)? Revisiting their study goal, they aimed at replicating that study using either a back shape reconstruction device. Along with this direct comparison they can give additional information concerning confounding parameters like heel-height or subjects’ age and BMI, which was more standardized in their own study.
Following that, the authors can compare their own results – or surface topography results in general – to those achieved with other instruments, and then it would be beneficial to report that radiographic and surface topography parameters are not directly comparable. And here, a note reflecting the non-invasive character of surface reconstruction would be beneficial in comparison to the radiation exposure methods of x-ray imaging.
The paragraph ‘Influence of precision of fiducial marker placement’ includes some method specific strengths and weaknesses: e.g. the manual positioning of the markers depends on the anatomical knowledge of the examiners. In contrast, video rasterstereography provides an automatically recognition of anatomy landmarks or geometrical inflectional points of the sagittal curvature in order to calculate spine shape parameters in real time. Thus, this paragraph should rather be the first part of the ‘limitations section’, to the reviewer’s opinion.
In its present form, the structure of the discussion appears to be to far from the study goal, to the reviewer’s opinion. I suggest a reorganization of the discussion section, but I am looking forward to reading your probable rebuttal.
Limitations of the study:
OK
Conclusions:
OK.

Author Response
We are thankful for the excellent critique pointing to the weaknesses of our manuscript. We do hope the reading of the reviewed version and our responses will be acceptable. Based on the comments, we have made the changes in the manuscript and responses are separately uploaded. The responses are highlighted in yellow.

Reviewer 3 Report
This study evaluates an interesting research question and obtained data from a large number of subjects initially. However, there are some significant methodological issues which will limit the utility of these findings. Below are some major and minor comments/suggestions:
Major comments:
Methodological issues:
Lines 213-215- Only 54 subjects out of 76 had full data. This indicates a major problem with the methods of data collection. Then another 9 were excluded as outliers. Only 56% of participants ended up being included. This means that almost half of the potential data was thrown out which is not an acceptable amoun The authors do little to convince me that this is a reliable methodology. There is a lengthy subjective description in the methods of why they believe they are accurate with the marker placement, but the only reference cited there is not a high quality reference. No reliability, SEM, or MDC data is presented Some major methodology remain unclear- for example, in Figure 3- at what angle are these being taken? It seems odd to look at VBA, a sagittal plane motion, from an oblique angle.Readability:
There are many grammatical errors throughout which decrease readability The discussion section needs significant work. First of all, the discussion is far too long. Secondly, the discussion should not just provide a literature review of one study and then another. It would be useful to synthesize the information and present studies that had common findings or methodology together. Most importantly, you need to compare and contrast your results with the findings from other studies- otherwise this is a literature review and not a discussion for your study. If the results are not relevant with respect to your findings, it would be best not to include that study There are a lot of acronyms that make the paper difficult to follow. I would suggest writing out things like “SHS” and “VBA” since there are so many of themMinor comments:
Lines 26-27- “The anterior pelvic tilt, lumbar lordosis angle and sacral tilt differences were usually mentioned in studies where the effects were high.” Please reword this sentence as the meaning is unclear. Lines 37-38- Please consider rewording “Wearing high-heeled shoes belongs to women's social behavior” as this is unclear. Line 44- What about “lower extremities?” Please specify- is this kinematics, kinetics? Line 49- Risk of what kind of injury? Falls? Orthopedic injury? Lines 60-62- If a specific study is going to be built upon, it would be appropriate to better highlight this study in the introduction and to better explain the study limitations which you will be addressing. Why is there a need for a more homogenous group? What information does this add to the previous knowledge? Was the problem the homogeneity of the group or the study methods (which you highlight in the next sentence). What about their sample size- was it inadequate in some way? If a previous study is being partially reproduced, the authors need to better explain the original study and need for more research. Line 70- Please define the acronym BMI during the first use (rather than the second). After BMi has been defined, the acronym can be used rather than writing both the term and the acronym Lines 69-75- While there is a lengthy description of the appropriate BMI, it does not specify if BMI >29 was an exclusion criterion (which would be appropriate given the information provided here). Line 116- Why does Figure 2 come before Figure 1 in the text? Lines 108-115- Why were these landmarks selected as opposed to bony landmarks more often used such as acromion (as opposed to axilla which has a lot of soft tissue) or PSIS as opposed to the dimple and the intergluteal furro (which is clearly influenced by gluteal tissue). Can you justify these landmarks? Lines 169-177- Are there known MDC or SEM measures for these variables? It might be useful for the reader to better understand what kinds of differences are real differences as opposed to just measurement error Lines 222-224- Just to be clearer, I would specify that these were differences between the barefoot and high heel conditions Lines 245-246- I do not understand why medium heels would have higher stability than low (if that is what is being implied). What is meant by “but which is smaller than the base of the heel support” Line 249-250- It says that they used 60mm but then that they used a variety of heel heights- which is it? Line 290-291- Repeatability does depend on the measurement uncertainty of the device, but even more so it depends on the repeatability of your marker placement- was reliability of this assessed? Lines 375-376- It is unclear what is meant by this sentence Lines 388-389- “Induce vertical integration” is not commonly used terminology and is unclear Line 402-408- This is completely subjective- do you have objective data that your measurements are reliable? This would be especially important given that almost half of your subjects were not included due to methodological problems. Reference 126 listed is not an adequate reference source on which to base this entire study.Author Response
We are thankful for the relevant critique pointing to the weaknesses of our manuscript. We do hope the reading of the reviewed version and our responses will be acceptable. Based on the comments, we have made the changes in the manuscript and responses are separately uploaded. The responses are highlighted in yellow.

Round 2
Reviewer 2 Report
Dear authors,
thank you very much for your efforts in the revised manuscript.
My concerns have been addressed completely.
I appreciate your modifications and updated statistical calculations, and I a satisfied with your responses to reviewers recommendations or your rebuttal.
Thank you very much and congratulations for your intersting manuscript.
Reviewer 3 Report
Thank you for your answers. Thank you also for including the ICC for the barefoot measurements as this does improve the manuscript.
I continue to believe that the use of a data set with only 54 or 76 subjects is a major limitation. Overall, despite a lot of work by the authors, I continue to struggle with the language and writing in this manuscript.